# Safety profile of inactivated COVID-19 in healthy adults aged ≥ 18 years: A passive surveillance in Indonesia

Nastiti Kaswandani[1,2☯], Bernie Endyarni Medise[1,2☯], Elcha Leonard[2☯]*, Hindra Irawan Satari[1,2‡], Julitasari Sundoro[2‡], Sri Rezeki Harun Hadinegoro[1,2‡], Ade Putra[2], Putra Fajar Angkasa[2]

1 Faculty of Medicine, Department of Child Health, University of Indonesia, Central Jakarta, DKI Jakarta, Indonesia, 2 National Adverse Events Following Immunization (NC AEFI) of Indonesia, Central Jakarta, DKI Jakarta, Indonesia

☯ These authors contributed equally to this work.
‡ These authors also contributed equally to this work.
* elchalnrd@gmail.com

## Abstract

Coronavirus Disease 2019 (COVID-19) vaccination in Indonesia has shown effectiveness in reducing the morbidity and mortality of Covid-19. The study aims to evaluate the incidence rate and severity of Adverse Events Following Immunization (AEFI) of inactivated SARS-CoV-2 vaccine during the first quarter of 2021 until the second quarter of 2022 in Indonesia. More than two hundred million Sinovac/CoronaVac were given from January 13th, 2021, until June 30th, 2022. Data for this study were collected manually and electronically from the national vaccine safety website managed by the National Committee (NC) of AEFI Indonesia and the Ministry of Health Indonesia. The total number of injections observed in the study was 264,311,992 doses consisting of 142,449,795 (first dose), 121,613,324 (second dose), and 248,873 (booster dose). Of the injections given, 301 subjects with Serious AEFIs (SAE) and 10.261 subjects with non-serious AEFIs (AE) reported, with a majority of SAE and AEs found in the first dose. Most of the SAEs were classified as coincidental events by the NC AEFI (IR 0.8/1 million doses on first dose injection; 0.31 on second dose injection). ISRR (immunization stress-related response) is in the second rank of SAEs reported (0.59 IR/1 million doses on the first dose; 0.14 on the second dose). The incidence rate of SAEs and AEs, both in the variable of age, sex, and symptoms per 1 million dose injections in Indonesia, was very rare according to WHO guidelines. Most SAEs were classified as coincidences or unrelated to the vaccine. The result showed that the Sinovac/CoronaVac in Indonesia is safe.

## Introduction

Coronavirus disease 2019 (COVID-19) is a new emerging disease first discovered in Wuhan in December 2019 [1,2]. Since its first emergence, WHO has reported 762,792,152 confirmed

**Data Availability Statement:** All relevant data are within the manuscript and its Supporting Information files.

**Funding:** This study was supported by a grant from Sinovac Ltd. The sponsor had no role in the study design, data collection, analysis or interpretation of data, writing of the report or decision to submit the article for publication NK BEM EL HIS JS SRH AP PFA.

**Competing interests:** The authors have declared that no competing interests exist.

COVID-19 cases and 6,897,025 deaths worldwide [3]. As of April 2023, there are 6,752,606 confirmed cases of COVID-19, with 161,071 deaths in Indonesia, reported to WHO [3]. Indonesia has the highest number of confirmed cases and deaths from COVID-19 in Southeast Asia and ranks second in Asia after India [4]. The world is still trying to control the spread of COVID-19 through lockdowns in several regions and mass vaccination programs [5].

Indonesia has initiated a COVID-19 vaccination program starting in January 2021. The priority population vaccinated was the frontline healthcare workers and public service workers, which gradually expanded to older people, all adults and children over six years old, all over Indonesia. Improving the inoculation rate of the COVID-19 vaccine to achieve the national herd immunity goal is essential [6].

As of June 2022, the total COVID-19 vaccine in Indonesia was 420,536,944 doses from various types. Sinovac/CoronaVac is the majority vaccine used in Indonesia (264,311,992 doses/ 62.8%) compared to other vaccines such as AstraZeneca/Vaxzveria (70,997,367/16.9%), Pfizer/ Comirnaty (65,759,501/15.6%) Moderna/mRNA-1273/Spikevax (14,937,124/ 3.6%), and Sinopharm/BBIBP-CorV (4,530,960/1.0%) [7]. The Sinovac/Coronavac was chosen due to its availability during the vaccination period and accessible cold chain considering the condition of healthcare facilities in 34 provinces in Indonesia. Although the safety profiles for Sinovac/CoronaVac have been well-investigated and qualified through various vaccinations, the adverse events may vary depending on the region and ethnicity. Also, there is a possibility of unreported side effects in the study trial [8–11]. In addition, several adverse events associated with COVID-19 vaccines have been reported, and disinformation through the mass media has been the cause of considerable anxiety among people about vaccine safety [6,11,12]. Therefore, it is essential to investigate, evaluate and analyze the occurrence of adverse reactions after vaccination in different regions and populations [6,8].

The National Committee of AEFI Indonesia (NC AEFI) has been actively contributing to ensuring the safety of all vaccines used in Indonesia since 1998, including COVID-19 vaccines. In this study, we conducted research based on the manual AEFIs report from health facilities and web reporting system in every province in Indonesia (passive surveillance) to assess the prevalence and characteristics of adverse reactions following the inactive COVID-19 vaccine (Sinovac) used in Indonesia (January 13th, 2021 until June 30th, 2022).

## Objectives

This study evaluated the incidence and severity of AEFI associated with the COVID-19 vaccine in Indonesia from the first quarter of 2021 to the second quarter of 2022 (January 13th, 2021 – June 30th, 2022). Despite the geographical challenges and varied economic background, we want to show the passive surveillance data resulting from Indonesia's established vaccine safety monitoring. We also want to describe the real-world data following the safety profile of inactivated SARS-CoV-2 vaccine during the national vaccination program against COVID-19 in Indonesia.

## Material and methods

A cross-sectional descriptive study was conducted from January 13th, 2021, to June 30th, 2022. The population of this study is all of the healthy adults in Indonesia who had experienced AEFI and reported it to the healthcare facility after completing the first, second, or booster doses of Sinovac/CoronaVac. The data was collected using passive surveillance via the manual reporting system and vaccine safety website developed by NC AEFI and the Health Ministry of Indonesia, established in 2015. This vaccine safety website reporting system used a webpage to document every report of AEFIs systematically. Based on the Ministry of Health Decree, every

healthcare facility with an immunization service was obligated to report serious AEFIs (SAEs) and non-serious AEFIs among the vaccinated people to the vaccine safety website. The healthcare facility should notify SAEs report in less than 24 hours, followed by the local health district for investigation and the Local Committee of AEFI and NC AEFI for causality assessment. Data were entered and cleaned using Microsoft Excel. Descriptive frequencies and incidence rate statistics were utilized to describe the status of AEFI with COVID-19.

The ethical clearance of this study has been obtained from the Faculty of Medicine, University of Indonesia, with No KET-41/UN2.F1/ETIK/PPM.00.02/2023. Since the collected data is secondary data owned by the NC-AEFI and there is no disclosing of the subjects' private information, written informed consent was not needed for this study.

## Operational definitions

1. Serious AEFI (SAE) is defined as every AEFI needing hospitalization, severe damage, permanent disability, and even death [13–15].

2. Non-serious AEFI (AE) is every medical symptom experienced after the immunization that was not requiring any specific medical treatment and the following observation. Local AEs symptoms included tenderness, pain at rest, redness, and swelling at the injection site [13–15]. AE systemic symptoms consist of fatigue, headache, malaise, arthralgia, chills, fever, etc. [13,15]

3. The AEFI surveillance from the health facilities or vaccination site will report the AEFI case (serious and non-serious)[15]. Free-text reporting allowed the health facilities, public health office, and local AEFI committee to describe any other symptoms and record the type and amount of pain medications used in the vaccinated patients. The serious AEFIs will be followed by the investigation by the public health office and causality assessment by the Local Committee of AEFI and the National Committee of AEFI according to the WHO algorithm [15].

4. The solicited symptom is the expected signs and symptoms after vaccination. Based on the previous phase three clinical trial in Indonesia, the solicited symptoms of Sinovac/CoronaVac consist of local and systemic reactions [16]. The solicited local reactions were local pain, redness, swelling, and induration. The systemic reactions were fever, myalgia, and fatigue [16].

5. The unsolicited symptom is the sign and symptoms after vaccination which is not expected to show in the study trial, thus, in real-world data [16].

6. Immunization Stress Related Response (ISRR) is used to describe a range of signs and symptoms that are related to "anxiety" around immunization and are not necessarily directly related to the vaccine product, a defect in the quality of the vaccine, or an error of the immunization program [17–19]. ISRR can manifest as an acute stress response, vasovagal reaction, or dissociative neurological symptom reaction (DNSR) with no apparent physiological basis. The onset of ISRR may take hours to days after immunization [17,18]

1. Indeterminate is used to classify SAEs that are consistent in a temporal relationship, but there is insufficient definitive evidence for vaccine causing the event (maybe a new vaccine-linked event) or if there are any other qualifying factors resulting in conflicting trends of consistency and inconsistency with causal association to immunization [13,14]

2. Unclassifiable is used to specify the additional information required for classification [13,14].

**Table 1. Total dose, SAE, and AE of SINOVAC COVID-19 vaccine reported in Indonesia up to June 30[th,] 2022.**

|  | Total Dose | Total Serious AEFI reported (n cases) | Total AEFI reported (n cases) |
|---|---|---|---|
| **First Dose** | 142,449,795 | 239 | 7,958 |
| **Second Dose** | 121,613,324 | 62 | 2,184 |
| **Booster Dose** | 248,873 | - | 74 |
| **Total** | 264,311,992 | 301 | 10,216 |

## Results

A total of 264.311.992 doses were given from January 13[th,] 2021, until June 30[th,] 2022, consisting of 142.449.795 (first dose injection), 121.613.324 (second dose injection), and 248.873 (booster dose). (Table 1) All provinces in Indonesia had reported the SAE and AE on the vaccine safety website, as shown in Fig 1.

There were 301 SAEs and 10,216 AEs reported on the vaccine safety website, whereas most cases occurred in the first dose injection. The incidence rate of SAEs per 1 million doses was 1.68 on the first dose\ and 0.51 on the second. The incidence rate of AEs per 1 million doses was 55.86 on the first dose and 17.96 on the second (Table 2).

Out of 301 SAEs reported, the most frequently reported cases in the age group are in the 31–45 (84; 0.61 IR/1 million dose) for the first and second doses (17; 0.14 IR/1 million doses). The female group reported more cases than men in the first dose (152 vs 87) and second dose injection (35 vs 27).

According to the WHO algorithm, all of the SAEs in this study were investigated and assessed by the NC-AEFI of Indonesia. Most SAEs were coincidental events in the first or second dose (114;0.8 IR/1 million dose and 38;0.31 IR/1 million dose), meaning most reported SAEs were unrelated to the vaccine. Additionally, ISRR (immunization stress-related response) is the second rank of SAEs classification in the first and second doses (84;0.59 IR/1 million doses and 17;0.14 IR/1 million doses) (Table 2).

Ten thousand two hundred sixteen subjects with AEs were reported in the first, second, and booster doses. Most of the SAEs occurred in the first dose. The booster doses were

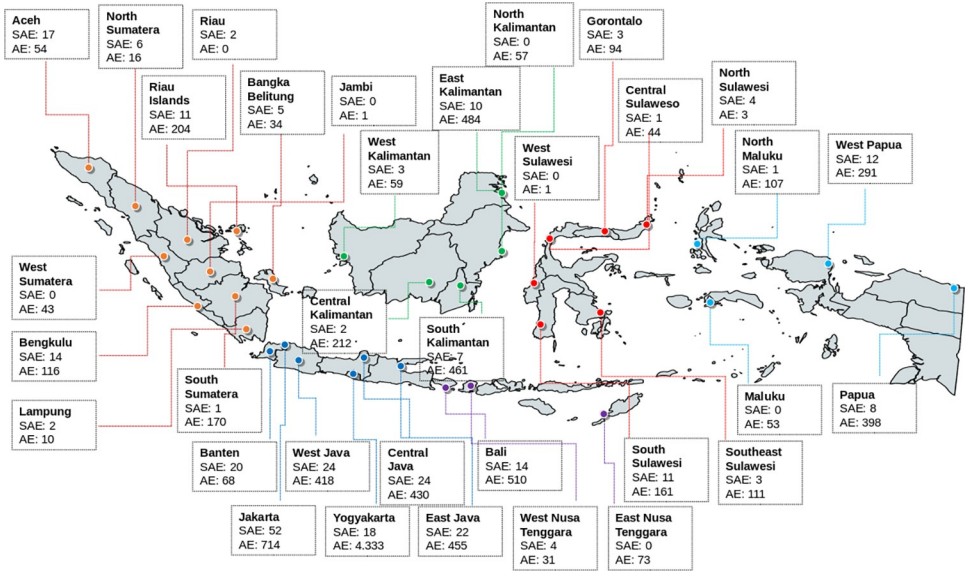

**Fig 1. Distribution of SAE and AEs reported by the 34 provinces in Indonesia.**

**Table 2. Subject's characteristics with serious AEFIs.**

| Variables | Vaccine Dose, n cases (IR per 1 million doses) | |
|---|---|---|
| | **1st Dose** | **2nd dose** |
| **Age** | | |
| **18–30** | 66 (0.46) | 15 (0.12) |
| **31–45** | 87(0.61) | 17 (0.14) |
| **46–59** | 38 (0.27) | 14 (0.12) |
| **>59** | 48 (0.34) | 16 (0.13) |
| **Gender** | | |
| **Male** | 87 (NA) | 27 (NA) |
| **Female** | 152 (NA) | 35 (NA) |
| **Classification** | | |
| **Vaccine Reaction** | 28 (0.20) | 4 (0.03) |
| **ISRR[a]** | 84 (0.59) | 17 (0.14) |
| **Coincidental events** | 114 (0.8) | 38 (0.31) |
| **Indeterminate** | 4 (0.03) | 2 (0.02) |
| **Unclassifiable** | 9 (0.06) | 1 (0.01) |

[a]ISRR: Immunization Stress Related Response.

presented in Table 3 as total cases because the total booster doses were less than one million injections, and the incidence rate counts could not be performed.

Similar to the SAEs, the most frequently reported AE in the age group variable is in the 31–45 for the first and second doses (22.31 and 7.73 IR/1 million doses). In the sex group, females reported more cases than men in the first dose (5,104 vs 2,770) and second dose (1,391 vs 766) (Table 3).

Table 4 shows that the subject's systemic symptoms were more experienced in the first (7697;54.03 IR/1 million doses) and second dose (1948;16.02 IR/1 million doses) than local symptoms. The table also shows unsolicited symptoms were more frequently reported than solicited symptoms in the first or second doses.

All SAE symptoms were documented on the vaccine safety website and shown in Fig 2. The most frequently reported in the first dose injections were nausea/vomitus, malaise/fatigue, and fever (0.46; 0.40; 0.39 IR/1 million doses). Meanwhile, in the second dose were headache, chest palpitation, and cough/rhinitis/sore throat (0.20; 0.18;0.18 IR/1 million doses).

**Table 3. Subjects' characteristics with non-serious AEFIs.**

| Variables | Vaccine Dose, n cases (IR per 1 million doses) | | |
|---|---|---|---|
| | **1st Dose** | **2nd dose** | **Booster dose** |
| **Age** | | | |
| **18–30** | 2,912 (20.44) | 773 (6.36) | 19[a] |
| **31–45** | 3,178 (22.31) | 940 (7.73) | 43[a] |
| **46–59** | 1,536 (10.78) | 381 (3.13) | 11[a] |
| **>59** | 332 (2.33) | 90 (0.74) | 1[a] |
| **Gender** | | | |
| **Male** | 2,770 (NA) | 766 (NA) | 21[a] |
| **Female** | 5,104 (NA) | 1,391 (NA) | 53[a] |

[a]AEFI rate is not performed in the booster dose injection because the total vaccines given were less than 1 million doses.

**Table 4. Solicited and unsolicited symptoms in total AEFIs.**

| Reaction | 1st dose n Events (IR /1 million dose) | | 2nd dose n Events (IR /1 million dose) | | Booster Dose n Events, (IR /1 million dose)[a] | | TOTAL[b] |
|---|---|---|---|---|---|---|---|
| | Local | Systemic | Local | Systemic | Local | Systemic | |
| **Solicited** | 2440 (17.13) | 3108 (21.82) | 814 (6.69) | 651 (5.35) | 53[a] | 57[a] | 7013 (26.52)[b] |
| **Unsolicited** | 83 (0.58) | 7697 (54.03) | 16 (0.13) | 1948 (16.02) | 0[a] | 63[a] | 9744 (36.87)[b] |

[a]AEFI rate is not performed in the booster dose injection because the total vaccines given were less than 1 million injection.

[b]In total reaction, we did not incorporate the local and systemic symptoms in the booster dose to get the incidence rate of the solicited and solicited symptoms in the first and second doses.

Fig 3 illustrates the incidence rate symptoms of SAEs Sinovac/CoronaVac based on the interval time. The bar chart shows that SAEs mainly occurred in the first dose more than 30 minutes after injection.

The overall incidence rate symptoms of SAEs are described in Fig 4. In the first doses, the most frequently reported symptoms were drowsiness (12.87 IR/1 million doses), fever (11.27 IR/1 million doses), and local pain (11.06 IR/1 million doses). Meanwhile, the most frequently reported symptoms in the second dose injections were local pain (4.49 IR/1 million doses), drowsiness (4.06 IR/1 million doses), and nausea/vomitus (2.68 IR/1 million doses).

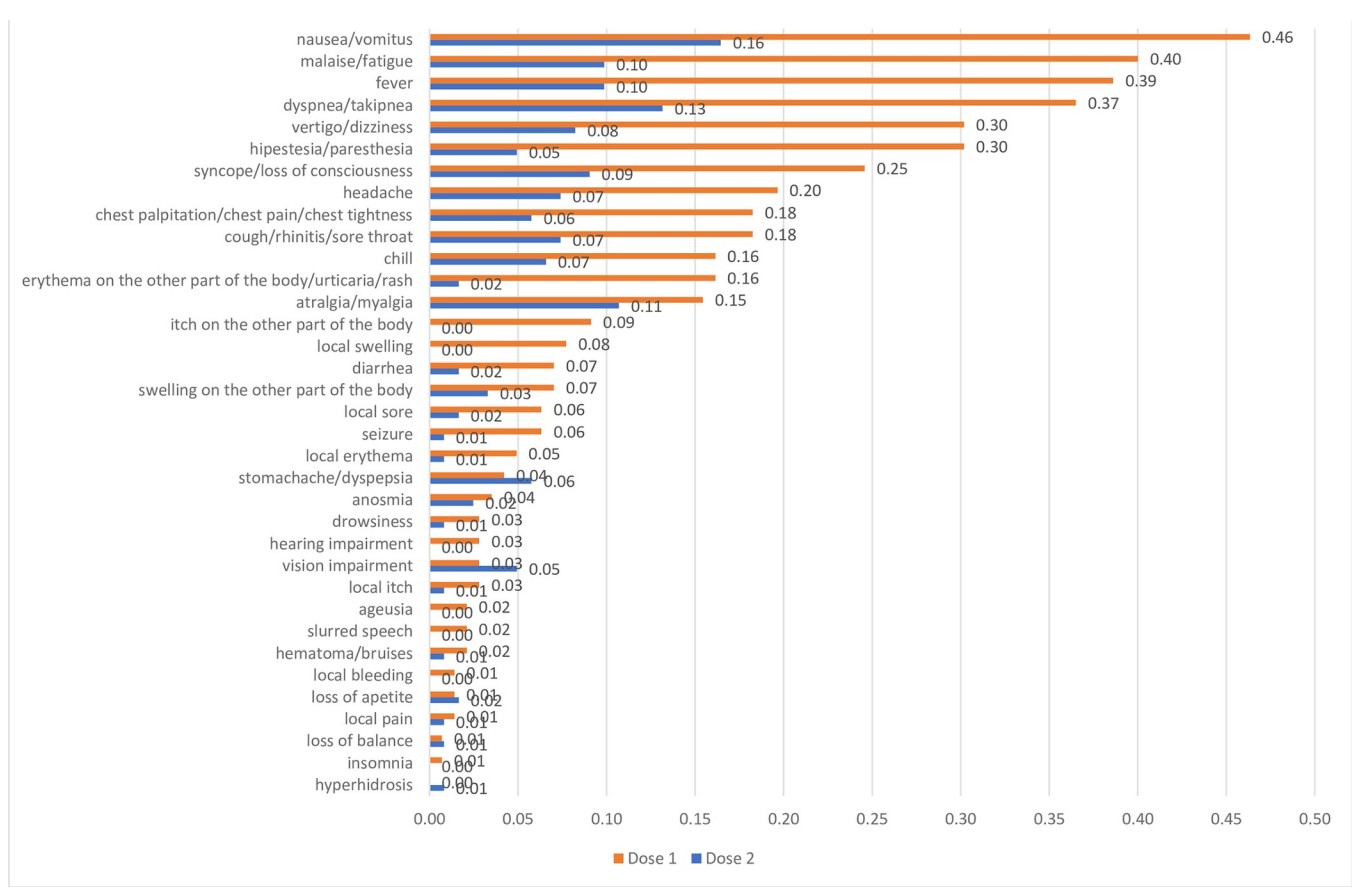

**Fig 2. Incidence Rate/1 M doses symptoms of SAE Sinovac/CoronaVac reported in vaccine safety website.**

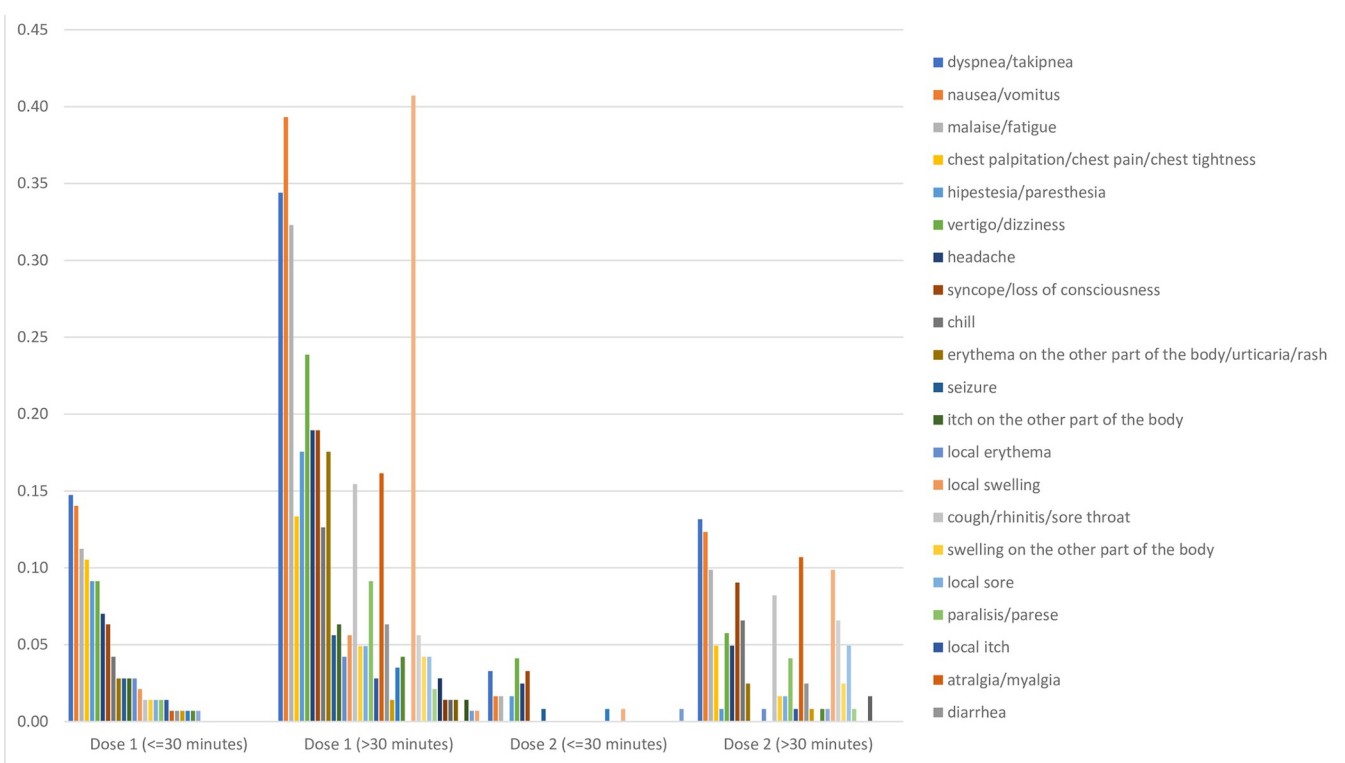

**Fig 3. SAEs symptoms based on time interval.**

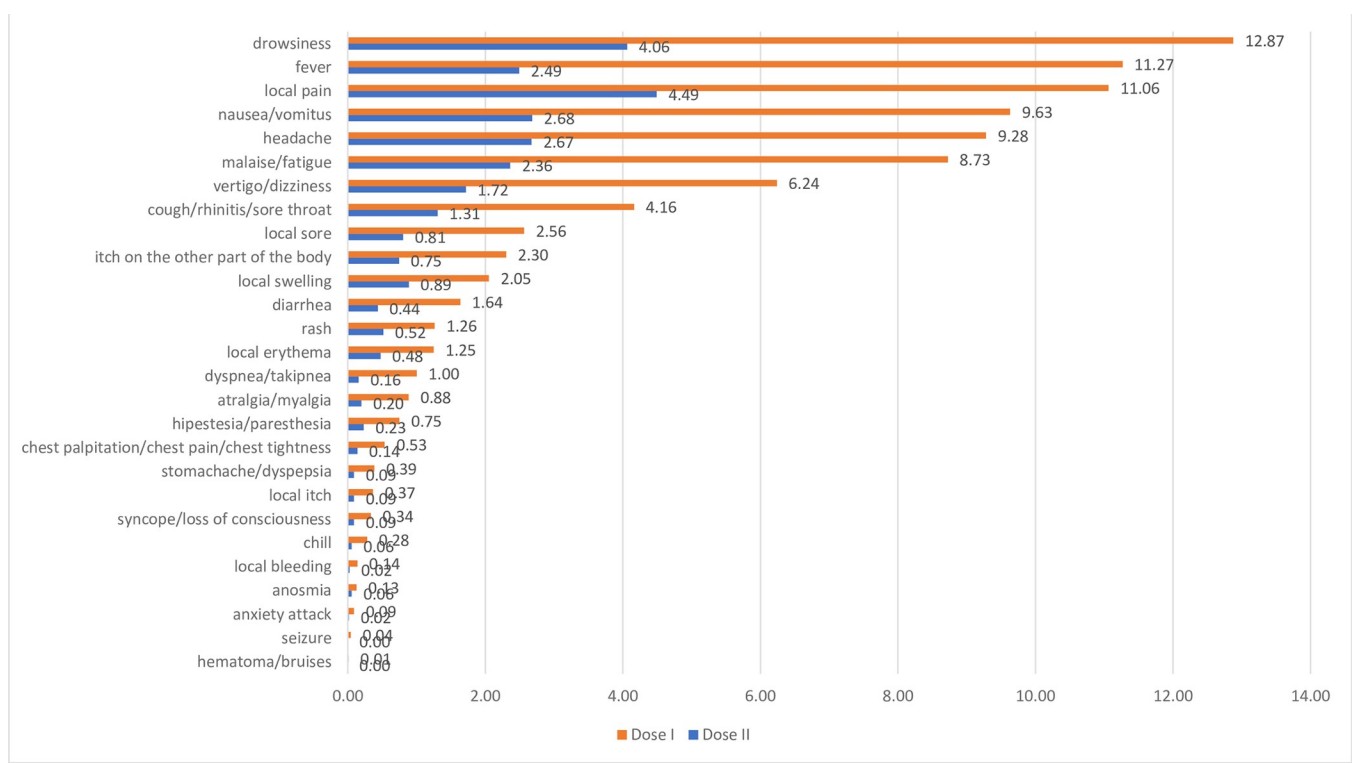

**Fig 4. Incidence Rate/1 M doses symptoms of AE Sinovac/CoronaVac reported on vaccine safety website.**

## Discussion

There were 301 SAEs and 10,216 AEs reported on Indonesia's vaccine safety website from January 13th, 2021—June 30th, 2022. We calculated and analyzed the SAE/AE incidence rate per 1 million doses of the demographic groups.

All SAEs in this study have been investigated, and causality assessed by the local and NC AEFI. Most reported cases were classified as coincidental events in the first and second dose (114; 0.8 and 38;0.31 IR/1 million dose) and thus considered inconsistent with the immunization [20]. It is proven that the most severe symptoms experienced after immunization reported in Indonesia are caused by other health events in the vaccine recipients.

Of 301 serious AEFIs, 84 SAEs (0.59 IR/1 million doses on first dose) and 17 SAEs (0.23 IR/1 million doses on second dose injections) were classified as ISRR. Our finding supports the data from Thailand that most reported AEFIs in the study were classified as ISRR. These results also aligned with the WHO guideline that ISRRs tend to occur when new vaccines are introduced or when changes to an established vaccination program, such as a new age group or new vaccination setting [19,21].

In SAEs and AEs, most cases frequently reported were experienced by females rather than males. Unfortunately, we could not elaborate on the incidence rate per gender group since no specific national data for Sinovac/Coronavac was provided by gender. Our data showed similar results to Duijster et al., which found that the incidence of any AEFI, local reactions, and the top 10 most frequent AEFIs were higher in females than males for both doses of four different platforms of the COVID-19 vaccine. [22] These were most likely to happen because most females mount more robust antibody responses against vaccination and experience more distinctive symptoms after immunization compared to males, which is mainly attributable to levels of sex hormones and chromosomal differences [23].

In the age group, the most frequently reported AEs in the variable age group are in the 31–45 for the first and second doses (3178;22.31 and 940;7.73 IR/1 million doses). Our study also showed that the elderly group had less AEFI reported in SAEs or AEs compared to another age group. These findings were similar to the active surveillance done in Southeast Brazil, resulting in mild and moderate symptoms for older people without related SAEs. [21]. PAHO report showed identical results that AEs mainly occur in the age group of 30 to 39 in America [24]. Wang et al. also stated equivalent results that there was a higher incidence of local and systemic AEs in young people than in the elderly (OR = 1.10, 95% CI (1.08, 1.12), P<0.01; OR = 1.18, 95% CI (1.14, 1.22), P<0.01) [25]. The reason is the reduced activity of thymocytes and thymic epithelial cells and immune response substances in the elderly, causing the immune function to decline [25]. Therefore, young adults are more prone to experience AE compared to the elderly, proving that inactivated COVID-19 vaccine is also safe for the elderly. In the trial phases 1, 2, and 3 done in several countries, the data also showed the adverse events reactions were considered safe and well tolerated in the child and adults in the immunocompromised group (HIV, autoimmune and elderly) [26–34].

In our study, adverse reactions occurred predominantly after the first dose injection. This result was aligned with the study phase 3 trial in Brazil, showing that adverse reactions tended to be less frequent with the second dose [10].

We found that the solicited adverse events were most likely to happen in the local reaction. In contrast, the unsolicited adverse events were most likely in the systemic reaction in the first and second doses. Our study showed that the SAEs group's most common adverse reactions were nausea/vomitus, malaise/fatigue, and fever. Meanwhile, the most frequent symptoms reported were drowsiness, local pain, and nausea/vomitus in the AEFIs group. These findings were notable since unsolicited systemic symptom (nausea/vomitus) was the highest prevalence

in SAE symptoms. We also found drowsiness (unsolicited systemic reaction) as the most occurred symptom in the AEs group. Benjamanukul et al. also found similar symptoms following the injection of Sinovac/CoronaVac, such as nausea, diarrhea, hypersomnia, fatigue, hunger, rash, and dizziness, which aligned with our study [26]. These results have proven the importance of continuous vaccine safety surveillance to substantiate and justify the previous results from the clinical phase trial with real-world data. With this purpose, all of the symptoms occurring after the vaccine can be observed whether later in the future it will become predictable to happen following the immunization (solicited symptoms).

Our study observed no reported SAEs in the booster dose of Sinovac/CoronaVac, and fewer reported AEFIs compared to the first and second doses. Pan et al. found a similar result that the severity of solicited local and systemic adverse reactions reported within 28 days after the third dose was lower than the previous dose (grade 1 to grade 2 in all vaccination cohorts) [32].

Overall, the data shows that the incidence rate of SAEs and AEs both in the variable of age, sex, and symptoms per 1 million doses were infrequent compared to the total of the dose given (IR <1/10.000) based on the WHO guidelines [19]. Therefore, the Sinovac/CoronaVac is considered as safe.

The strength of this study is that this is one of the largest scales of Covid-19 mass vaccinations in the world, supported by manual and electronic national report systems. The participants in this study were from different risk groups nationwide, rendering the results more generalizable to the real-world context. Additionally, all SAEs have been investigated and assessed to determine the causality by the NC AEFI, an independent national committee consisting of various experts according to the WHO algorithm, giving the study a new perspective to the researcher worldwide.

This study also has limitations. The data were collected with a passive surveillance method; hence there is a probability of unreported AEFIs that were not analyzed in this study. Also, we could not elaborate on the incidence rate of SAE/AEs by gender due to data insufficiency.

## Conclusion

In conclusion, our data showed that the incidence rate of SAEs and AEs, both in the variable of age, gender, and symptoms per 1 (one) million doses, was very rare. Most of the SAEs were considered coincidence or not related to the vaccine. The result showed that the Sinovac/CoronaVac in Indonesia is safe to use. The monitoring and reporting of AEFI should be done continuously to ensure vaccine safety in Indonesia.

## Supporting information

**S1 File.**
(ZIP)

## Acknowledgments

We want to express our deepest gratitude to the local committee of AEFI, the focal point of surveillance, and the local health district, who have contributed to the data collection and SAEs assessment.

## Author Contributions

**Conceptualization:** Nastiti Kaswandani, Bernie Endyarni Medise, Elcha Leonard, Hindra Irawan Satari.

**Data curation:** Nastiti Kaswandani, Bernie Endyarni Medise, Elcha Leonard, Ade Putra, Putra Fajar Angkasa.

**Formal analysis:** Nastiti Kaswandani, Elcha Leonard, Ade Putra, Putra Fajar Angkasa.

**Investigation:** Elcha Leonard.

**Methodology:** Nastiti Kaswandani, Bernie Endyarni Medise, Elcha Leonard, Hindra Irawan Satari, Julitasari Sundoro, Sri Rezeki Harun Hadinegoro.

**Project administration:** Elcha Leonard.

**Resources:** Elcha Leonard.

**Supervision:** Nastiti Kaswandani, Bernie Endyarni Medise, Hindra Irawan Satari, Julitasari Sundoro, Sri Rezeki Harun Hadinegoro.

**Validation:** Nastiti Kaswandani, Bernie Endyarni Medise, Sri Rezeki Harun Hadinegoro.

**Visualization:** Nastiti Kaswandani, Bernie Endyarni Medise, Elcha Leonard.

**Writing – original draft:** Nastiti Kaswandani, Bernie Endyarni Medise, Elcha Leonard.

**Writing – review & editing:** Nastiti Kaswandani, Bernie Endyarni Medise, Elcha Leonard, Hindra Irawan Satari, Julitasari Sundoro, Sri Rezeki Harun Hadinegoro.

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
