## [Decision Letter · Decision Letter 0]

7 Mar 2023

PONE-D-23-02102Safety Profile of An Inactivated SARS-CoV-2 Vaccine (Sinovac/CoronaVac) in Healthy Adults aged ≥ 18 Years: a Passive Surveillance in IndonesiaPLOS ONE

Dear Dr. Leonard,

Thank you for submitting your manuscript to PLOS ONE. After careful consideration, we feel that it has merit but does not fully meet PLOS ONE’s publication criteria as it currently stands. Therefore, we invite you to submit a revised version of the manuscript that addresses the points raised during the review process. Particularly, the study's primary strength is its use of appropriate statistical analysis of national surveillance data. Nevertheless, its weak aspect is the insufficient depth in analyzing and discussing the study's findings. In addition, stating the exact percentage of the Sinovac COVID-19 vaccine administered is crucial, rather than merely indicating that it constituted the majority of the vaccine utilized. Finally, the discussion section should be more comprehensive and in-depth.

We look forward to receiving your revised manuscript.

Kind regards,

Mohammad-Reza Malekpour

Academic Editor

PLOS ONE

Journal Requirements:

2. Please ensure that you have specified (1) whether consent was informed and (2) what type you obtained (for instance, written or verbal, and if verbal, how it was documented and witnessed). If your study included minors, state whether you obtained consent from parents or guardians. If the need for consent was waived by the ethics committee, please include this information.

"This study was supported by a grant from Sinovac Ltd. The sponsor had no role in the study design, data collection, analysis or interpretation of data, writing of the report or decision to submit the article for publication"

"No"

7. Please ensure that you include a title page within your main document. You should list all authors and all affiliations as per our author instructions and clearly indicate the corresponding author.

Reviewers' comments:

Reviewer's Responses to Questions

**Comments to the Author**

1. Is the manuscript technically sound, and do the data support the conclusions?

Reviewer #1: Yes

Reviewer #2: Yes

Reviewer #3: Yes

2. Has the statistical analysis been performed appropriately and rigorously? 

Reviewer #1: No

Reviewer #2: Yes

Reviewer #3: Yes

3. Have the authors made all data underlying the findings in their manuscript fully available?

Reviewer #1: No

Reviewer #2: Yes

Reviewer #3: No

4. Is the manuscript presented in an intelligible fashion and written in standard English?

Reviewer #1: No

Reviewer #2: No

Reviewer #3: Yes

5. Review Comments to the Author

Reviewer #1: The study is aimed to investigate the rate of adverse reactions of COVID-19 vaccination in Indonesia to decrease the vaccination hesitancy and be a reliable source for Indonesian people's bodies' probable pathological response to the vaccine. I do agree that these kind of studies are required for each nation but there are some points that need to be considered first:

The manuscript needs intensive language editing. I suggest using language editing services.

Introduction, lines 21-24: the statistics needs to be updated.

Materials and methods: the first three subheadings can be merged.

In the study results it is reported that female gender was accounted for more adverse events after 1st, second, or booster dose. But is there any proof that females weren’t more vaccinated than males? For concluding that adverse events are more common among females, we are going to need such fractions: females having symptoms/total vaccinated females Vs. males having symptoms/total vaccinated males.

Generally, it is highly recommended to report the study population’s characteristics as a part of your results.

Reviewer #2: The manuscript by Nastiti et al looked into the Safety Profile of Inactivated Vaccine COVID-19 in Healthy Adults Aged above 18 years in Indonesia. The study is good but need following major revision

1. Too many grammatical errors throughout the manuscript. It required a through combing grammatical and spelling error preferably from an expert.

2. The abstract first line ‘…………is one of the methods to prevent…..’ may be reframed.

3. Sentence in line 52-55 not clear.

4. Line 55 ‘world’ not ‘word’.

5. Authors may consider adding the ‘Study design’ section briefly.

6. The sum of ‘Total Serious AEFI reported (n cases)’ in table 1 is not correct.

7. Fix line 143 and 144: 5.104 instead of 5,104 and 1.391 instead of 1,391.

8. Line 167 to 180: Figure numbering seems inconsistent

9. Line 176: remove , with . in data

Reviewer #3: 1. The manuscript reports on the incidence and severity of Adverse Events Following Immunization (AEFI) after the nationwide roll out of the Sinovac COVID-19 vaccine in Indonesia. The data consists of passive national surveillance developed by the government of Indonesia, and reported to a vaccine safety website and through manual data collection. Data for the study were collected between January 13th 2021 and 31st June 30 2022. The strength of the study lies in the analysis of national surveillance data and the use of appropriate statistics. However, the weakness of the study is in the lack of depth in the analysis and discussion of findings. I have recommended minor revisions .

2. Given that a major strength of the study is the use of the national surveillance data, the analysis should go beyond provides descriptive outputs about AEFI but include an assessment of the quality or completeness of the data. Authors mention that all health facilities are required by policy to report into the system, but there is no analysis to show if indeed all facilities providing the COVID-19 vaccine are reporting. This is important to inform the interpretation of the AEFI incidence rate presented in the analysis for the whole country.

3. It is important to report the proportion of COVID-19 vaccines in use that was Sinovac. It is not enough to say it was the majority of the vaccine used, since the paper positions this analysis as a study of AEFI associated with the use of Sinovac. If other vaccines were in use at the same time, please report them and their percentages.

4. The methods section does not describe their approach to analysis. Authors should revise this section and include what analysis was done and what statistically tests were done - if any. There should also be a statement around ethical review - if one was sought or done, if not why not.

5. The discussion section should do a bit more than report what was found, it should interpret the results and tell the reader the implications of the findings. For example, authors could discuss the AEFI rates in relation to expected rates or discuss any potential challenge with the data that could affect the inference drawn from their findings.

6. The manuscript could do with copy editing to clean out grammatical errors and fix punctuations.

6. PLOS authors have the option to publish the peer review history of their article (what does this mean?). If published, this will include your full peer review and any attached files.

Reviewer #1: **Yes: **Mohammadreza Azangou-Khyavy

Reviewer #2: No

Reviewer #3: **Yes: **Dr. Chizoba Barbara Wonodi

---

## [Author Response · Author response to Decision Letter 0]

28 Apr 2023

Mohammad-Reza Malekpour, M.D.

Academic Editor

Plos One

April 18th 2023

Dear Academic Editor, Dr. Malekpour:

Subject: Submission of Revised Paper Safety Profile of An Inactivated SARS-CoV-2 Vaccine (Sinovac/CoronaVac) in Healthy Adults aged ≥ 18 Years: a Passive Surveillance in Indonesia PLOS ONE [PONE-D-23-02102] - [EMID:fb8ac8d3d8d39754]

We are grateful for your kind and generous feedback in the email dated March 8th, 2022 enclosing the reviewer's comments. Your comments provided valuable insights to refine its contents and analysis. We have carefully reviewed the comments and have revised our manuscript accordingly. Our responses are listed below. Changes to the manuscripts are shown in the highlight or with track changes in the document.

Thank you for giving us the opportunity to submit a revised draft of the manuscript "Safety Profile of An Inactivated SARS-CoV-2 Vaccine (Sinovac/CoronaVac) in Healthy Adults aged ≥ 18 Years: a Passive Surveillance in Indonesia" for publication in the PlosOne. We sincerely hope the revised document will answer all the comments and be suitable for publication. We are looking forward to hearing from you in due course.

Sincerely yours,

Nastiti Kaswandani, M.D., Paediatrician, PhD

Bernie Endyarni Medise, M.D., Paediatrician, PhD

Elcha Leonard, M.D.

And researcher’s team

Responses to Reviewer 1 

- Introduction, lines 21-24: the statistics need to be updated.

As suggested by the reviewer, we have updated the latest statistical data on COVID-19 cases and deaths, and the changes can be found in lines 21-25 

- Materials and methods: the first three subheadings can be merged.

We have merged the first three subheadings in the material and methods section (lines 62-74)

- In the study results, it is reported that the female gender accounted for more adverse events after first, second, or booster dose. But is there any proof that females weren't more vaccinated than males? For concluding that adverse events are more common among females, we are going to need such fractions: females having symptoms/total vaccinated females Vs. Males having symptoms/total vaccinated males.

Thank you for this suggestion. It would have been interesting to explore this aspect. Unfortunately, this would not be possible in our study because there is no specific national data for total Sinovac/Coronavac doses provided by gender group.

Responses to Reviewer 2

- The abstract first line ‘…………is one of the methods to prevent…..’ may be reframed

We have reframed the first line in the abstract (lines 2-3)

- Sentence in lines 52-55 is not clear.

We agree with the reviewer's assessment. We have revised the suggested part in the manuscript to make it more knowledgeable to the readers. (lines 56-57) 

- Line 55' world' not 'word'

Thank you for pointing this out. The reviewer is correct, and we have changed it to "world"

- Authors may consider adding the 'Study design' section briefly.

Thank you for your suggestion. We have added the study design section and tried to merge it into the material and methods section. (lines 62-78)

- The sum of 'Total Serious AEFI reported (n cases)' in Table 1 is not correct.

We agree with the reviewer's assessment. Accordingly, we have revised the total Serious AEFIs reported (n cases) throughout the manuscript. 

- Fix lines 143 and 144: 5.104 instead of 5,104 and 1.391 instead of 1,391.

While we appreciate the reviewer's feedback, we respectfully disagree. The 5,104 and 1,391 stated in the manuscript mean to show the total number of cases of AE reported in female and males group, respectively.

- Line 176: remove , with . in data

Thank you for pointing this out. We have changed it according to the reviewer's suggestion.

Responses to Reviewer 3

- Given that a major strength of the study is the use of the national surveillance data, the analysis should go beyond provides descriptive outputs about AEFI but include an assessment of the quality or completeness of the data. Authors mention that all health facilities are required by policy to report into the system, but there is no analysis to show if indeed all facilities providing the COVID-19 vaccine are reporting. This is important to inform the interpretation of the AEFI incidence rate presented in the analysis for the whole country.

We think this is an excellent suggestion. We have portrayed the SAE/AE reports from all of the provinces in Indonesia in Figure 1. We have also added the total number of reported AEFIs from various vaccination centers in Indonesia, listed in Supplementary Files 2.

- It is important to report the proportion of COVID-19 vaccines in use that was Sinovac. If other vaccines were in use at the same time, please report them and their percentages.

We agreed with the reviewer's suggestion. We have added the total doses of COVID-19 vaccination given in Indonesia simultaneously with various platforms and their percentages (lines 33-36).

- The methods section does not describe their approach to analysis. Authors should revise this section and include what analysis was done and what statistically tests were done - if any. There should also be a statement around ethical review - if one was sought or done, if not why not.

As suggested by the reviewer, we have incorporated the analysis and statistical test that have been done. We have also added the ethical review of this study (lines 75-78)

- The discussion section should do a bit more than report what was found, it should interpret the results and tell the reader the implications of the findings. For example, authors could discuss the AEFI rates in relation to expected rates or discuss any potential challenge with the data that could affect the inference drawn from their findings.

Thank you for your exquisite opinion. We have tried refining the discussion section and further explaining potential challenges in this study.

- Grammatical errors and fix punctuation

Thank you for pointing this out. As the reviewers suggested, we have fixed this study's grammatical errors and punctuation.

---

## [Decision Letter · Decision Letter 1]

17 May 2023

Safety Profile of An Inactivated SARS-CoV-2 Vaccine (Sinovac/CoronaVac) in Healthy Adults aged ≥ 18 Years: a Passive Surveillance in Indonesia

PONE-D-23-02102R1

Dear Dr. Leonard,

We’re pleased to inform you that your manuscript has been judged scientifically suitable for publication and will be formally accepted for publication once it meets all outstanding technical requirements.

Kind regards,

Mohammad-Reza Malekpour

Academic Editor

PLOS ONE

Reviewers' comments:

Reviewer's Responses to Questions

**Comments to the Author**

1. If the authors have adequately addressed your comments raised in a previous round of review and you feel that this manuscript is now acceptable for publication, you may indicate that here to bypass the “Comments to the Author” section, enter your conflict of interest statement in the “Confidential to Editor” section, and submit your "Accept" recommendation.

Reviewer #1: (No Response)

Reviewer #2: All comments have been addressed

Reviewer #3: All comments have been addressed

2. Is the manuscript technically sound, and do the data support the conclusions?

Reviewer #1: Yes

Reviewer #2: Yes

Reviewer #3: Yes

3. Has the statistical analysis been performed appropriately and rigorously? 

Reviewer #1: Yes

Reviewer #2: Yes

Reviewer #3: Yes

4. Have the authors made all data underlying the findings in their manuscript fully available?

Reviewer #1: Yes

Reviewer #2: Yes

Reviewer #3: Yes

5. Is the manuscript presented in an intelligible fashion and written in standard English?

Reviewer #1: Yes

Reviewer #2: Yes

Reviewer #3: Yes

6. Review Comments to the Author

Reviewer #1: Regarding the previous comment about females experiencing higher side effects, if the authors don't have sufficient supporting data which I had asked, (as they have declared) they need to mention it in the discussion section.

Reviewer #2: (No Response)

Reviewer #3: (No Response)

7. PLOS authors have the option to publish the peer review history of their article (what does this mean?). If published, this will include your full peer review and any attached files.

Reviewer #1: **Yes: **Mohammadreza Azangou-Khyavy

Reviewer #2: No

Reviewer #3: No

---

## [Editor Report · Acceptance letter]

26 May 2023

PONE-D-23-02102R1 

Safety Profile of Inactivated COVID-19 in Healthy Adults Aged ≥ 18 years: A Passive Surveillance in Indonesia 

Dear Dr. Leonard:

I'm pleased to inform you that your manuscript has been deemed suitable for publication in PLOS ONE. Congratulations! Your manuscript is now with our production department. 

Kind regards, 

on behalf of

Dr. Mohammad-Reza Malekpour 

Academic Editor

PLOS ONE